# Dental Caries and the Erosive Tooth Wear Status of 12-Year-Old Children in Jakarta, Indonesia

**DOI:** 10.3390/ijerph16162994

**Published:** 2019-08-20

**Authors:** Diah Ayu Maharani, Shinan Zhang, Shiqian Sherry Gao, Chun-Hung Chu, Anton Rahardjo

**Affiliations:** 1Department of Preventive and Public Health Dentistry, Faculty of Dentistry, Universitas Indonesia, Jakarta 10430, Indonesia; 2Faculty of Stomatology, Kunming Medical University, Yunnan 650500, China; 3Faculty of Dentistry, The University of Hong Kong, Hong Kong SAR 999077, China

**Keywords:** dental caries, erosive tooth wear, oral health behaviors, epidemiology, children

## Abstract

Background: Indonesia has the largest population of all countries in southeast Asia. However, little information is available on the oral health status of Indonesian children. The aims of this study were to assess dental caries and erosive tooth wear in 12-year-old children in Jakarta, Indonesia and to investigate the associated risk factors. Methods: Samples were selected using cluster sampling. Parents were asked to complete a self-administered questionnaire regarding their oral health knowledge, demographic information, their child’s dietary habits, and oral health-related behaviors. Experience of caries and erosive tooth wear were recorded using the Decayed, Missing (due to caries), and Filled Teeth (DMFT) index and the Basic Erosive Wear Examination (BEWE) index, respectively. Results: Of 779 children invited, 696 participated in the survey. Of these, 61% had experienced caries, and the mean DMFT score was 1.58. Almost all decay was untreated. Children who were female, who had a high frequency of soft drink intake, and whose father’s educational level was low were more likely to have dental caries. Most children had at least one lesion of erosive tooth wear. Children whose mother’s educational level was low were more likely to have erosive tooth wear. Conclusions: The prevalence of dental caries and erosive tooth wear was high in 12-year-old children in Jakarta. Their dietary habits and parental level of education were associated with the presence of these dental conditions.

## 1. Introduction

Although public knowledge about and attitudes towards dental caries have improved in recent decades, it remains the most prevalent oral disease in children worldwide and historically, has been considered the most important global oral health burden [1]. If untreated, dental caries progress, causing pain and infection. In addition, they can have a significant impact on a child’s daily functions, including chewing, swallowing, and speaking, which can influence nutrient absorption, growth, and general health [2]. Studies have reported that the presence of dental caries is associated with a child’s poor oral health-related quality of life [3,4]. It is well accepted that the oral health related behaviors of children and their families’ socioeconomic background are associated with the presence of dental caries [5].

Dental erosion is defined as an irreversible loss of hard dental tissue due to the chemical influence of extrinsic acid, such as that from diet and medication, and of intrinsic acid, such as that from gastroesophageal reflux and vomiting, without bacterial involvement [6,7]. Erosive tooth wear is the subsequent loss of chemically-softened tooth tissue by an abrasive force [8]. It is a multifactorial condition involving the interplay of various chemical, biological, and behavioral factors. The erosive potential depends on chemical factors, including pH, titratable acidity, mineral content, and the calcium-chelating properties of the dental tissue. Biological factors, such as saliva, the acquired pellicle, tooth structure, and the position of a tooth in relation to the soft tissues and tongue, are associated with the pathogenesis of erosive tooth wear. Furthermore, behavioral factors, including eating and drinking habits, excessive oral hygiene, and emotional disorders, are predisposing factors for erosive dental wear [9].

Indonesia has the largest population of all countries in southeast Asia, and it is the fourth largest country in the world in terms of population. It consists of more than 1300 ethnic groups [10]. Jakarta, as the capital city of Indonesia, has a heterogeneous population comprising numerous ethnicities and socioeconomic strata and is representative of the Indonesian population [11]. The epidemiological examination of oral health status in a representative sample is important for monitoring the prevalence and severity of oral diseases and to plan and provide preventive measures as early as possible. World Health Organization (WHO) guidelines have recommended that an epidemiological survey should assess the oral health status of school children at 12 years of age [12], as all permanent teeth, except for third molars, should have erupted by this age [13,14]. However, little attention has been paid to the oral health status of children in Indonesia, on which there is limited data available in the literature [15,16,17,18]. Based on the background detailed above, the aims of this study were to assess dental caries and the erosive tooth wear status of 12-year-old children in Jakarta, Indonesia and to investigate the risk factors associated with dental caries and erosive tooth wear in these children.

## 2. Materials and Methods

The present study was performed in Jakarta, Indonesia in 2016. Ethics approval was obtained from the Ethics Review Board of the Faculty of Dentistry at Universitas Indonesia (6/Ethical Approval/FKGUI/VIII/2016). This study was reported according to the Strengthening the Reporting of Observational Studies in Epidemiology (STROBE) statement [19].

Sample size estimation was performed with reference to a previous survey of school children in Jakarta, which reported that the prevalence of decayed, missing, and filled teeth in 12-year-old children was 84% [20]. The marginal error of the estimate was set at ≤ 3%. With a two-sided confidence level (CI) of 95%, the computed minimum sample size was 574. Assuming a response rate of 85%, the number of participants to be invited needed to be at least 675. A cluster sampling method was used to select 32 schools from a list of secondary schools according to the population distribution of six geographical regions in Jakarta. This method resulted in the random selection of four schools in central Jakarta (95 children), six schools in west Jakarta (128 children), six schools in south Jakarta (134 children), eight schools in east Jakarta (186 children), seven schools in north Jakarta (126 children), and one school in the Thousand Islands (27 children), all of which agreed to participate. Children in these schools who were 12 years old were invited to join the study. Any children who did not cooperate with the oral health examination or who had severe general health problems were excluded. Written informed consent was obtained from the children’s parents.

Research assistants visited each school prior to the day of the examination to discuss the protocol with teachers and deliver the questionnaires to parents. A self-administered questionnaire, which has been used in a previous study, was used in this survey [21]. The questionnaire consisted of three sections: (1) information on the child’s demographic background (gender, age, and place of birth); (2) oral health-related behaviors, such as the frequency of tooth brushing and dietary habits (frequency of intake of soft drinks, citrus drinks, fruit juice, chewing gum, and vitamin C supplement drinks); and (3) parental oral health knowledge. To assess the oral health knowledge of parents, 21 multiple choice questions on the causes and prevention of oral health diseases were included in the questionnaire Appendix A. Each correct answer was given a score of 1, and an incorrect answer or “don’t know” was given a score of 0. Therefore, the total score for parents’ dental knowledge ranged between 0 and 21. Total scores were categorized into three groups representing parents’ dental knowledge: low (a score of 0–7), middle (a score of 8–14), and high (a score of 15–21).

The clinical examination was performed by a trained and calibrated dentist using a ball-ended Community Periodontal Index probe and disposable dental mirror attached to an intraoral light-emitting diode light. Duplicate examinations were performed on 10% of the children in each school visit. The kappa statistic was used to assess intra-examiner reproducibility. A diagnosis of dental caries in permanent teeth was made according to the criteria recommended by the WHO [12]. Caries were diagnosed when a lesion had an unmistakable cavity, a detectably softened floor or wall, or undermined enamel. The Decayed, Missing (due to caries), and Filled Teeth (DMFT) index was used to record caries in children’s permanent dentition. The Basic Erosive Wear Examination (BEWE) criterion was used to assess dental erosion status [22]. The buccal, lingual, and occlusal/incisal surfaces of all permanent teeth were examined. The severity of erosive tooth wear was recorded at four levels: (1) a score of “0” = no erosive tooth wear, (2) a score of “1” = initial loss of surface texture, (3) a score of “2” = distinct defect with hard tissue loss over < 50% of the surface area (dentine often involved), and (4) a score of “3” = hard tissue loss over ≥ 50% of the surface area (dentine often involved). The surface with the highest BEWE score in a sextant was recorded to represent the entire sextant.

Data from clinical examinations and questionnaires were entered into Excel files by two investigators. Data cleaning was performed prior to analysis. Data analysis was performed using Statistical Package for the Social Sciences, version 20 (IBM Corp., Inc., Chicago, IL, USA). Descriptive analysis was performed to determine the prevalence of dental caries and erosive tooth wear in participants. The chi-square test was used to assess the relationship between independent variables and the presence of dental caries and erosive tooth wear. Logistic regression analysis using a backward stepwise procedure was performed to evaluate the risk factors for dental caries and erosive tooth wear. The level of statistical significance for all tests was set at 0.05.

## 3. Results

A total of 779 children from 32 schools were invited to participate in the study. Of these, 83 failed to complete the questionnaire or the dental examination. Therefore, 696 children were included in the study. The response rate was 89%, of which 43% were male (*n* = 300), and the kappa values for intra-agreement assessment of dental caries and erosive tooth wear were 0.90 and 0.87, respectively.

### 3.1. Dental Caries Status

Children’s caries experience, assessed using the DMFT index (± SD), was 1.58 ± 2.03 (Table 1). Untreated decayed teeth (DT) contributed to 93% of the DMFT score (mean DT = 1.52 ± 1.99). Girls had a higher DMFT score than boys (1.70 ± 2.06 compared with 1.44 ± 1.98, *p* = 0.016). Girls also had more untreated DT than boys (1.60 ± 2.02 compared with 1.40 ± 1.96, *p* = 0.044). The proportion of children who had experienced dental caries (DMFT > 0) was 61% (*n* = 423). Among these, 263 (38%) had a low level of caries (DMFT score = 1 or 2), 99 (14%) had a moderate level of caries (DMFT score = 3 or 4), 36 (5%) had a high level of caries (DMFT score = 5 or 6), and 25 (4%) had a very high level of caries (DMFT score > 6) based on WHO criteria. The chi-square test revealed that gender; frequency of intake of soft drinks, citrus drinks, and chewing gum; and parental level of education were associated with the prevalence of dental caries (Table 2).

The results of logistic regression analysis revealed that the prevalence of dental caries was significantly associated with gender (odds ratio (OR): 1.528, 95% CI: 1.116–2.092, *p* = 0.008), frequency of soft drink intake (OR: 1.980, 95% CI: 1.365–2.871, *p* < 0.001), and fathers’ educational level (OR: 2.431, 95% CI: 1.322–4.471, *p* = 0.004; OR: 1.597, 95% CI: 1.105–2.308, *p* = 0.013; Table 3).

### 3.2. Dental Erosion Status

A total of 665 children (289 boys and 376 girls) had at least one erosive lesion. The prevalence of dental erosion was 96%. Most children (86%, 599/696) presented with a distinct defect with hard tissue loss of < 50% of the surface area (BEWE score = 2), and none had severe erosive tooth wear (BEWE score = 3, Table 4). The chi-square test results indicated that the frequency of soft drink intake, parental educational level, and parental dental knowledge were significantly associated with the presence of erosive tooth wear (Table 5). The results of logistic regression analysis are presented in Table 6. Children whose mothers’ educational level was low (OR: 14.303, 95% CI: 1.867–109.564, *p* = 0.010; OR: 6.066, 95% CI: 2.829–13.005, *p* < 0.001) were more likely to have erosive tooth wear.

## 4. Discussion

Despite the large population in Indonesia, epidemiological studies on the oral health status of children have been limited [15,16,17,18]. As the capital of Indonesia, Jakarta was considered a reference for other major cities in Indonesia. The total population in Jakarta is divided into six districts, each of which includes eight to 10 subdistricts. In the present study, 32 schools were selected randomly in each subdistrict, represented by 10–41 children from each school, based on district’s population proportion. The present study investigated dental caries and the erosive tooth wear status of 12-year-old school children in Jakarta, which was considered representative of the urban population in Indonesia. Socioeconomic status is an important risk factor for caries, so school children from six geographical districts in Jakarta were selected to represent its heterogeneous population [23]. However, there were no significant socioeconomic differences between the districts, which have relatively low levels of poverty compared to other areas in Indonesia [24].

The DMFT index is the most widely used method to record the experience of caries in permanent teeth. The present study diagnosed caries according to WHO criteria, which ensured that the results were comparable with those from other countries. Several indices are available to measure erosive tooth wear status [25], including the BEWE [22], the Smith and Knight Tooth Wear Index [26], and the Simplified Scoring Criteria for Tooth Wear Index [27]. However, there is little agreement in terms of which index should be used as the standardized method in epidemiological surveys. The BEWE index was introduced in 2007 and has been used to identify erosive tooth wear at the patient level. It has been used in epidemiological surveys worldwide [21,28]. In the present study, the BEWE index was used to assess erosive tooth wear, as it is a validated and simple instrument, which is less time-consuming and more practical than other measures [29]. In addition, by using a widely adopted instrument, the results of this study can be compared with those of others, which increases its impact.

The present study found that 61% of 12-year-old Indonesian children were affected by dental caries, and the mean DMFT score was 1.58. The prevalence of caries and DMFT score were higher than those in other regions of southeast Asian [21,30,31], which indicated that these Indonesian children had a poorer oral health status. In this study, girls were more likely to have dental caries. This was consistent with a widely documented argument that females have presented with a higher prevalence of dental caries than males throughout time and across cultures [32,33]. Teeth eruption usually occurs earlier in girls than in boys, and the longer period of exposure can lead to a longer duration of contact between teeth and a cariogenic oral environment [33]. The present study demonstrated that a high frequency of soft drink intake was a risk factor associated with dental caries in Indonesian children. It is well accepted that soft drinks can cause dental caries due to their high sugar content, which can be metabolized by oral biofilms to generate acid, which can lead to the demineralization of hard tissues and dental caries [34]. The findings of the present study confirmed the association between the intake of soft drinks and the presence of dental caries in school children. The survey also reported that children whose fathers were less well educated were more likely to have dental caries. In Indonesia, males are the dominant family members. It may be that the children of fathers with low education levels may have unhealthy dietary habits, which can be associated with dental caries. This result supported the fact that educational level is a key factor for caries worldwide [35].

The present study revealed that almost all participants had erosive tooth wear, and the prevalence was higher compared with that in other countries [21,36,37,38]. This difference may be attributed to dietary behaviors and socioeconomic factors. A higher frequency of soft drink intake can be a risk factor associated with erosive tooth wear. Theories have stated that the extrinsic etiology of dental erosion is associated with exposure to acidic substances; thus, more frequent exposure of the teeth to acidic substance increases their erosion [39]. The results of the chi-square test revealed that fathers’ and mothers’ education levels were associated with the prevalence of erosive tooth wear. In addition, the dental knowledge of parents was associated with the presence of erosive tooth wear in children. The level of education and dental knowledge of parents can affect their daily life decisions about their child’s dietary intake of acid. Therefore, less knowledge and a lack of education can increase the risk of children suffering from erosive tooth wear [21].

It is crucial to identify oral health problems in children and the risk factors associated with these problems to ensure the effective planning and delivery of subsequent interventions. Based on the global goals for oral health 2020 [40], it is essential to evaluate the current oral health situation and set adequate oral health goals, objectives, and targets. One target is to reduce the DMFT score, especially the D component, of 12-year-olds [40]. The present study found that dental caries and erosive tooth wear were prevalent in 12-year-old Indonesian children. Almost all dental caries were untreated, which may be due to inequity in the use of dental care services. A previous study reported that those living in remote areas of Indonesia had insufficient access to dental care due to the unequal distribution of dental care services between urban and rural areas [41]. However, children recruited to the present study lived in the capital city of Indonesia; therefore, these children should have had sufficient access to oral health care. Furthermore, basic dental services by a general dentist are included in universal health coverage [24]. Despite this, dental healthcare utilization is still relatively low, and children still have poor oral health [42]. Currently, patients seek dental care for pain rather than using it as a regular preventive measure [43].

A limitation of this study is that data on the clinical consequences of untreated dental caries, such as pulpal involvement, ulceration caused by dislocated tooth fragments, fistula and abscess (PUFA) were not collected. As there is a low care index in this group, this aspect would be interesting, as it is almost inevitable that within some years, it would result in acute treatment need (for pain, abscess, and so on) [44]. In the present study, most parents claimed that their children brushed their teeth at least twice or more per day; however, no relationship between oral hygiene behaviors and dental caries was found. Therefore, parental education and dental knowledge, which can influence a child’s dietary habits, may be the most important factors related to the compromised oral health of Indonesian school children. Oral health programs for children and parents should be developed and implemented according to evidence-based dentistry [45]. Currently, there is insufficient evidence for the efficacy of oral health programs for reducing caries in Indonesia. Further research should be conducted to evaluate potential interventions to improve the oral health of Indonesian children.

## 5. Conclusions

In conclusion, more than 50% of 12-year-old children in the present study had dental caries, which were mostly untreated. A high frequency of soft drink intake, fathers’ educational attainment, and being female were associated with a higher prevalence of caries. Many children had signs of erosive tooth wear, which was associated with their mothers’ level of educational attainment.

## Figures and Tables

**Table 1 ijerph-16-02994-t001:** Caries experience measured by Decayed, Missing (due to caries), and Filled Teeth (DMFT) index.

DMFT	Mean ± SD
Caries experience (DMFT)	1.58 ± 2.03
Decayed teeth (DT)	1.52 ± 1.99
Missing teeth due to caries (MT)	0.01 ± 0.11
Filled teeth (FT)	0.06 ± 0.32

**Table 2 ijerph-16-02994-t002:** Mean DMFT and caries prevalence according to selected variables.

Variable (*n*)	Mean ± SD DMFT	DMFT > 0 (*n*, %)	*p*-Value ^#^
Gender (696)			0.010 *
Female (396)	1.7 ± 2.1	256 (65%)	
Male (300)	1.4 ± 1.9	167 (56%)	
Place of birth (696)			0.497
Jakarta (552)	1.6 ± 2.0	336 (61%)	
Not Jakarta (144)	1.7 ± 2.1	87 (60%)	
Frequency of soft drinks (696)			0.001 *
At least once per week (185)	2.1 ± 2.2	134 (72%)	
Less than once per week (511)	1.4 ± 1.9	289 (57%)	
Frequency of citrus tea/drinks containing lemon (696)			0.037 *
At least once per week (415)	1.7 ± 2.1	264 (64%)	
Less than once per week (281)	1.4 ± 1.9	159 (57%)	
Frequency of fruit juice (696)			0.522
At least once per week (444)	1.6 ± 1.9	264 (59%)	
Less than once per week (252)	1.6 ± 2.1	159 (63%)	
Frequency of chewing gum (696)			0.012 *
At least once per week (251)	1.8 ± 2.2	167 (67%)	
Less than once per week (445)	1.5 ± 1.9	256 (58%)	
Frequency of vitamin C supplement drinks (696)			0.352
At least once per week (268)	1.6 ± 2.1	160 (60%)	
Less than once per week (428)	1.6 ± 2.0	263 (61%)	
Frequency of tooth brushing (696)			0.353
At least once a day (79)	1.5 ± 1.8	46 (58%)	
Twice or more (617)	1.6 ± 2.1	377 (61%)	
Caregiver (696)			0.457
Parents (653)	1.6 ± 2.1	396 (61%)	
Others (43)	1.4 ± 1.6	27 (63%)	
Education of father (696)			0.004 *
Primary or lower (73)	2.2 ± 2.3	53 (73%)	
Secondary (462)	1.6 ± 2.1	288 (62%)	
Tertiary or higher (161)	1.2 ± 1.7	82 (51%)	
Education of mother (696)			0.042 *
Primary or lower (99)	1.8 ± 1.9	66 (67%)	
Secondary (469)	1.6 ± 2.1	291 (62%)	
Tertiary or higher (128)	1.2 ± 1.9	66 (52%)	
Parental dental knowledge (696)			0.100
Low (66)	1.8 ± 2.2	42 (64%)	
Middle (497)	1.7 ± 2.1	311 (63%)	
High (133)	1.1 ± 1.6	70 (53%)	
Erosive tooth wear experience (696)			0.614
No (31)	1.2 ± 1.4	17 (55%)	
Yes (665)	1.6 ± 2.1	406 (61%)	

^#^ Chi-square test. * *p*-value < 0.05. DMFT, Decayed, Missing, Filled (due to caries) Teeth.

**Table 3 ijerph-16-02994-t003:** Results of logistic regression of the association between caries prevalence and explanatory variables among children in Jakarta (*n* = 696).

Variable	Odds Ratio	95% CI	*p*-Value
Gender Female Male ^#^	1.528	1.116–2.092	0.008
Education of father Primary or below Secondary Tertiary or above ^#^	2.4311.597	1.322–4.471 1.105–2.308	0.004 0.013
Frequency of soft drinks At least once per week Less than once per week ^#^	1.980	1.365–2.871	0.001

^#^ Reference group; DMFT, Decayed, Missing, Filled (due to caries) Teeth; CI, confidence interval.

**Table 4 ijerph-16-02994-t004:** Prevalence and severity of erosive tooth wear.

Gender	*n*	Prevalence (BEWE > 0)	Highest BEWE Score			
			0	1	2	3
Male	300	289 (96%)	11 (4%)	38 (13%)	251 (84%)	0 (0%)
Female	396	376 (95%)	20 (5%)	28 (7%)	348 (88%)	0 (0%)
Total	696	665 (96%)	31 (5%)	66 (10%)	599 (86%)	0 (0%)

BEWE, Basic Erosive Wear Examination.

**Table 5 ijerph-16-02994-t005:** Erosive tooth wear and studied variables.

Variable (*n*)	BEWE > 0 (*n*, %)	*p*-Value *
Gender (696)		0.381
Girls (396)	376 (95%)	
Boys (300)	289 (96%)	
Place of birth (696)		0.472
Jakarta (552)	529 (96%)	
Not Jakarta (144)	136 (94%)	
Frequency of soft drinks (696)		0.029 *
At least once per week (185)	182 (98%)	
Less than once per week (511)	483 (95%)	
Frequency of citrus tea/drinks containing lemon (696)		0.093
At least once per week (415)	401 (97%)	
Less than once per week (281)	264 (94%)	
Frequency of fruit juice (696)		0.932
At least once per week (444)	424 (96%)	
Less than once per week (252)	241 (96%)	
Frequency of chewing gum (696)		0.404
At least once per week (251)	242 (96%)	
Less than once per week (445)	423 (95%)	
Frequency of vitamin C supplement drinks (696)		0.724
At least once per week (268)	257 (96%)	
Less than once per week (428)	408 (95%)	
Frequency of tooth brushing (696)		0.780
At least once a day (79)	75 (95%)	
Twice or more (617)	590 (96%)	
Caregiver (696)		0.948
Parents (653)	624 (96%)	
Others (43)	41 (95%)	
Education of father (696)		0.001 *
Primary or lower (73)	72 (99%)	
Secondary (462)	452 (98%)	
Tertiary or higher (161)	141 (88%)	
Education of mother (696)		0.001 *
Primary or lower (99)	98 (99%)	
Secondary (469)	457 (97%)	
Tertiary or higher (128)	110 (86%)	
Parental dental knowledge (696)		0.001 *
Low (66)	66 (100%)	
Middle (497)	480 (97%)	
High (133)	119 (90%)	
Caries experience (696)		0.489
No (271)	259 (95%)	
Yes (423)	406 (96%)	

* Chi-square test, BEWE, Basic Erosive Wear Examination.

**Table 6 ijerph-16-02994-t006:** Results of logistic regression of the association between erosive tooth wear and explanatory variables among children in Jakarta (*n* = 696).

Variable	Odds Ratio	95% CI	*p*-Value
Frequency of soft drinks At least once per week Less than once per week ^#^	3.091	0.914–10.451	0.069
Education of mother Primary or lower Secondary Tertiary or higher ^#^	14.303 6.066	1.867–109.564 2.829–13.005	0.010 0.001

^#^ Reference group. BEWE, Basic Erosive Wear Examination; CI, confidence interval.

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
