# Peer review of "Dental Caries and the Erosive Tooth Wear Status of 12-Year-Old Children in Jakarta, Indonesia"

_ijerph, 2019, doi:10.3390/ijerph16162994_

Round 1

Reviewer 1 Report

The paper presents a relevant theme. I suggest describing how the parents' knowledge questionnaire was

Author Response

The Questionnaire was attached as supplementary file accordingly.

Reviewer 2 Report

Congratulations to a good piece of work in the field of epidemiology regarding oral health in children.

Nonetheless, I have some minor remarks mostly regarding format and the way of presenting data in the tables, that it is easier to read, but also regarding some other aspects in the discussion.

The introduction and M & M section are well written and comprise the most relevant or enough information.

but please,

In l. 61 & l.162 You write: “Despite the large population in Indonesia, epidemiological studies on the oral health status of  children have been limited.” Does it mean there are none? If there are citations would be good in introduction and discussion.

Table 1. you write “Decay teeth (DT) “-> it should be Decayed teeth (DT)

Regarding format and readability:

Table 2. for the reader it would be nice that aspects which are statistically significantly different are marked or highlighted in some way, then one doesn’t have to read all p-values in detail. I would put the data with % into brackets -> like you did it in table 4. -> better readability

I would suggest also to add an extra column for the mean DMFT of each variable in table 2.

Moreso, the headings in table 2 like sex, place of birth etc. should be clearly differentiated by its subpoints (e.g. female, male) either by font or position.

The same for table 3, 5, 6

Table 4 . why extra line under “male    300”? otherwise it is better to read and understand than the other tables.

Why don’t you give OR for all variables? -> I think it would be possible to add extra columns in table 2 and add the data from table 3. Similarly table 6 may be included to table 5. -> then you show more data but dont need more space

l. 173 what about "BEWE18"

I think it would be good to describe in one or two sentences whether there are socioeconomic differences between the 6 districts in Jakarta (either in Mat & Meth. or in the discussion part), as it is known that socioeconomic status is an important risk factor for caries

L.196 “This finding contradicts those reported in other cultures [17].” What exactly is contradictive? To my knowledge educational level is worldwide a key factor for caries. -> e.g. Schwendicke F, Dörfer CE, Schlattmann P, Foster Page L, Thomson WM, Paris S. Socioeconomic inequality and caries: a systematic review and meta-analysis. J Dent Res. 2015 Jan;94(1):10-8. doi: 10.1177/0022034514557546

In the discussion a more detailed explanation regarding the aspect of the low care index would be good. Why do you think did these children not receive dental care, despite that “these children should have sufficient access to oral health care” ? What are the costs for dental treatment in Indonesia in relation to costs of living? Is dental care covered by insurance, or do you have to pay for it privately? What is a typical pattern for dental visits? Do children usually visit the dentist only pain related?

l. 224 You write: “School-based oral health education programmes for children and parents should be developed and implemented.” This is possibly a good idea - but what exactly should these programmes include to be efficient in your country? And what is the status quo? Where do you see most potential for improvement?

In the conclusion (l.228) you write:

“ A high frequency of soft drink intake, fathers’ educational attainment and being female were found to be associated with the prevalence of caries.” -> I would suggest to write “associated with a higher prevalence of caries.

Some more ideas for the discussion:

How about putting the caries experience into relation with the WHO goals for 2020? https://www.who.int/oral_health/media/en/orh_goals_2020.pdf

Did you collect any data on pain/symptoms like PUFA? As there is a low care index in this group this aspect would be interesting on top, as it is almost inevitable that if not at the age of 12 but whitin some years it will result in acute treatment need (pain, abscess etc.).

Author Response

Congratulations to a good piece of work in the field of epidemiology regarding oral health in children. Nonetheless, I have some minor remarks mostly regarding format and the way of presenting data in the tables, that it is easier to read, but also regarding some other aspects in the discussion. The introduction and M & M section are well written and comprise the most relevant or enough information. But please, in l. 161 & l.162 you write: “Despite the large population in Indonesia, epidemiological studies on the oral health status of children have been limited.” Does it mean there are none? If there are citations would be good in introduction and discussion.

Response: Citations were added in the introduction and discussion section (line 63 and 166 & reference 15−18).

Table 1. You write “Decay teeth (DT) “-> it should be Decayed teeth (DT)

Response: Revision was made.

Regarding format and readability: Table 2. For the reader it would be nice that aspects which are statistically significantly different are marked or highlighted in some way, then one doesn’t have to read all p-values in detail.

Response: The statistically significant p-values were highlighted by bold texts and asterisk.

Table 2. I would put the data with % into brackets -> like you did it in table 4. -> Better readability

Response: Data with % in table 2 were put into brackets as suggested.

I would suggest also to add an extra column for the mean DMFT of each variable in table 2.

Response: An extra column was inserted in table 2 to describe the mean and standard deviation of the DMFT index per variable.

More so, the headings in table 2 like sex, place of birth etc. should be clearly differentiated by its sub points (e.g. female, male) either by font or position. The same for table 3, 5, 6.

Response: The tables were revised accordingly.

Table 4. Why extra line under “male 300”? Otherwise it is better to read and understand than the other tables.

Response: Table 4 was revised.

Why don’t you give OR for all variables? -> I think it would be possible to add extra columns in table 2 and add the data from table 3. Similarly table 6 may be included to table 5. -> Then you show more data but don’t need more space

Response: Table 2 and 5 are descriptive and bivariate analysis results. Whereas table 3 and 6 are results (end model) of the multivariable logistic regressions analysis. Therefor tables were distinct to improve readability.

173 what about "BEWE18"

Response: This was revised by inserting brackets for the number, because it was the reference (line 179).

I think it would be good to describe in one or two sentences whether there are socioeconomic differences between the 6 districts in Jakarta (either in Mat & Meth. or in the discussion part), as it is known that socioeconomic status is an important risk factor for caries

Response: Two sentences were added in the discussion section to address this issue and related references were inserted (line 172−175 & reference 23 and 24).

L.196 “This finding contradicts those reported in other cultures [17].” What exactly is contradictive? To my knowledge educational level is worldwide a key factor for caries. -> e.g. Schwendicke F, Dörfer CE, Schlattmann P, Foster Page L, Thomson WM, Paris S. Socioeconomic inequality and caries: a systematic review and meta-analysis. J Dent Res. 2015 Jan;94(1):10-8. doi: 10.1177/0022034514557546

Response: This sentence was revised and reference was added as suggested (line 203 & reference 35).

In the discussion a more detailed explanation regarding the aspect of the low care index would be good. Why do you think did these children not receive dental care, despite that “these children should have sufficient access to oral health care”? What are the costs for dental treatment in Indonesia in relation to costs of living? Is dental care covered by insurance, or do you have to pay for it privately? What is a typical pattern for dental visits? Do children usually visit the dentist only pain related?

Response: Further information was added in to describe the needful information with additional references (line 226−229 & reference 24, 42, 43).

224 You write: “School-based oral health education programmes for children and parents should be developed and implemented.” This is possibly a good idea - but what exactly should these programmes include to be efficient in your country? And what is the status quo? Where do you see most potential for improvement?

Response: The discussion section was added to address these issues (line 238−241 & reference 45).

In the conclusion (l.228) you write: “A high frequency of soft drink intake, fathers’ educational attainment and being female were found to be associated with the prevalence of caries.” -> I would suggest to write “associated with a higher prevalence of caries.

Response: The sentence was revised as suggested (line 245).

Some more ideas for the discussion: How about putting the caries experience into relation with the WHO goals for 2020? https://www.who.int/oral_health/media/en/orh_goals_2020.pdf

Response: As suggested, the discussion section was added with new reference accordingly (line 217−220 & reference 40).

Did you collect any data on pain/symptoms like PUFA? As there is a low care index in this group this aspect would be interesting on top, as it is almost inevitable that if not at the age of 12 but within some years it will result in acute treatment need (pain, abscess etc.).

Response: The discussion section was addressed accordingly (line 230−234 & reference 44).